# SOFIA: Selection of Medical Features by Induced Alterations in Numeric Labels

**Franklin Parrales Bravo** [1,2,*] **, Alberto A. Del Barrio García** [1] **, Luis M. S. Russo** [3] **and Jose L. Ayala** [1]

[1] Faculty of Computer Science, Complutense University of Madrid, Av. Séneca, 2, 28040 Madrid, Spain; abarriog@ucm.es (A.A.D.B.G.); jayala@ucm.es (J.L.A.)

[2] Carrera de Ing. en Sistemas Computacionales, Facultad Ciencias Matemáticas y Física, Universidad de Guayaquil, Guayaquil 090313, Ecuador

[3] INESC-ID and the Department of Computer Science and Engineering, Instituto Superior Técnico, Universidade de Lisboa, 1649-004 Lisboa, Portugal; luis.russo@tecnico.ulisboa.pt

* Correspondence: fparrale@ucm.es

**Abstract:** This work deals with the improvement of multi-target prediction models through a proposed optimization called Selection Of medical Features by Induced Alterations in numeric labels (SOFIA). This method performs a data transformation when: (1) weighting the features, (2) performing small perturbations on numeric labels and (3) selecting the features that are relevant in the trained multi-target prediction models. With the purpose of decreasing the computational cost in the SOFIA method, we consider those multi-objective optimization metaheuristics that support parallelization. In this sense, we propose an extension of the Natural Optimization (NO) approach for Simulated Annealing to support a multi-objective (MO) optimization. This proposed extension, called MONO, and some multiobjective evolutionary algorithms (MOEAs) are considered when performing the SOFIA method to improve prediction models in a multi-stage migraine treatment. This work also considers the adaptation of these metaheuristics to run on GPUs for accelerating the exploration of a larger space of solutions and improving results at the same time. The obtained results show that accuracies close to 88% are obtained with the MONO metaheuristic when employing eight threads and when running on a GPU. In addition, training times have been decreased from more than 8 h to less than 45 min when running the algorithms on a GPU. Besides, classification models trained with the SOFIA method only require 15 medical features or fewer to predict treatment responses. All in all, the methods proposed in this work remarkably improve the accuracy of multi-target prediction models for the OnabotulinumtoxinA (BoNT-A) treatment, while selecting those relevant features that allow us to know in advance the response to every stage of the treatment.

**Keywords:** multi-target classification; multi-objective optimization; GPU; feature weighting; feature selection

---

## 1. Introduction

Nowadays, the medical field has benefited from taking advantage of new technologies through the use of electronic medical records (EMRs). In fact, modern hospitals possess a wide variety of monitoring and data collection devices that provide relatively inexpensive means of collecting and storing data in inter and intrahospital information systems [1]. Hence, medicine has leveraged the availability of these EMRs to build predictive models for a given treatment [2]. Due to the economic cost of continuous treatments such as those of hepatitis C [3,4] or migraine [5], it is important for our methodology to be able to know in advance whether a treatment will be effective in patients with

those diseases. This can be achieved when predicting the response of all stages of treatment at the same time (multi-target prediction).

Beyond the technological benefits of EMRs, there are some challenges to solve before learning predictive models. One of the most common problems found in clinical datasets is the existence of many medical features and a low number of registers [6]. To solve that, some computational methodologies are applied to reduce the number of features, selecting only those that are relevant to a single class attribute (the feature to be predicted) in clinical datasets. For example, Armañanzas et al. [7] use the Feature Subset Selection (FSS) technique to reveal the most important factors when predicting the severity of symptoms in patients with Parkinson's disease [8] using non-motor symptoms. In other works [2,9], the authors propose a data transformation for carrying out the selection of relevant features while achieving high percentages of accuracy when predicting treatment responses in migraine. In these works, a single-objective Simulated Annealing (SA) [10] was applied to find that data transformation instead of employing a multi-objective optimization metaheuristic [9,11]. Moreover, SA has a high computational cost [12]. To solve that, this work considers the parallelization of SA through multithreading and GPUs to distribute the computational load on different cores of the computer while achieving high results at the same time [13]. Consequently, on the basis of the aforementioned works, it is desirable to extend these feature selection techniques to take into account various class attributes as in multi-target prediction models. In this way, it will be possible to select those relevant medical features considering all the responses to be predicted while finding that data transformation that allows to obtain high accuracies when performing a multi-target prediction.

Another aspect to solve is the heterogeneity of clinical data. Indeed, clinical data can come in the form of images (X-rays, magnetic resonance scans, etc.), interviews with the patients, laboratory data, as well as the doctor's observations and interpretations [14]. The homogeneity of the information can be addressed by simplifying and categorizing the data. For instance, this can be carried out through the transformation of heterogeneous clinical data to labels [15]. That labels need to be defined and agreed by the experts in the disease to be analyzed for achieving an adequate representation of the medical information [16]. Hence, as pointed out in the aforementioned works, it is desirable that this work should deal with heterogeneous data, leveraging the labelling provided by medical experts.

In this article, the Selection Of medical Features by Induced Alterations in numeric labels (SOFIA) method is presented. It is a methodology that allows to predict responses to all stages of a continuous treatment while selecting those relevant medical features that are relevant for knowing whether a treatment will be effective in patients. The SOFIA contributions can thus be summarized as follows:

- It considers the improvement of multi-target prediction models, allowing to know in advance whether a multi-stage treatment will be effective when no session has yet been made. The data transformation that leads to a high prediction percentage (optimal solution) has been efficiently explored by means of the SOFIA method, which improves the results expected by the use of the unmodified datasets when training prediction models.
- A multi-objective extension of the natural optimization (NO) approach for SA has been considered with a single-objective NO. For the proposed Multi-Objective Natural Optimization (MONO), the use of the hypervolume metric has been considered to accept those promising solutions. Moreover, MONO considers a parallel execution of the multi-objective optimization with the purpose of diminishing the computational cost [17].
- The proposed SOFIA methodology is applied in a realistic scenario when analyzing EMRs of migraineurs under the OnabotulinumtoxinA (BoNT-A) treatment. In this way, the multi-target prediction models have benefited significantly when performing the SOFIA method, achieving mean accuracies close to 88%. In addition, the selected medical features achieved through the SOFIA method have reduced the economic cost associated to collect all the medical features from a patient, allowing us to focus only on those that allow to know in advance the response that the treatment will have in the patients.

The rest of the paper is organized as follows. Section 2 describes the work related with some techniques applied to migraine and other illnesses. In Section 3, our methodology for predicting treatment results is explained. Section 4 presents the importance of applying the proposed methodology to predict multiple responses of migraineurs under the OnabotulinumtoxinA (BoNT-A) treatment. Section 5 describes the experiments and comparisons between different multi-objective metaheuristic methods when performing the SOFIA method. Moreover, the selected clinical features are presented and analyzed. Finally, our conclusions and future lines of work are presented in Section 6.

## 2. Related Work

Nowadays, some medical domains have leveraged the prediction of several clinical outcomes of patients at the same time [18]. In this sense, different machine learning methodologies are proposed for processing the EMRs and performing the simultaneous prediction of multiple target variables of diverse type. For example, in [19], a generalized framework to predict cognitive and symptomatic scores for schizophrenia and healthy controls using magnetic resonance imaging (MRI) is proposed. Moreover, in [20], the authors describe some computational tools for the prediction of chemical multi-target profiles and adverse outcomes with systems toxicology. In addition, in [21], a multi-layer multi-target regression is proposed for the prediction of cognitive assessment from multiple neuroimaging biomarkers, allowing an early detection of Alzheimer's disease. Finally, in the SMURF methodology [9], a panoramic prediction is contemplated for predicting the responses of a multi-stage treatment for the migraine. That approach is based on the improvement of multi-target classification models to give a general idea of how the patient will evolve after receiving certain medication.

In the aforementioned SMURF methodology [9], a data transformation that improves the accuracy percentages in multi-target prediction models is presented. This transformation consist of the multiplication between a feature weighting vector and the numeric labels of the columns in a clinical dataset. In addition, that transformation considers a rounding operation in order to induce alterations that help to select those features that are relevant to the class attributes (the features to be predicted) in the dataset. However, that technique, called "SAR encoding", is based on the use of the single-objective version of the SA metaheuristic called NO [22], and a rounding operation in order to find the optimal data transformation, those that achieves the best average of all accuracies in multi-target prediction models. In this sense, SOFIA can benefit from the use of multi-objective metaheuristics to improve all prediction accuracies at the same time instead of optimizing only the average accuracy in multi-target prediction models.

Due the promising accuracy results achieved in [9] when performing SA for a single-objective optimization, the NO method can be extended to the multi-objective scenario, allowing a parallel execution to reduce the computational cost [17]. In addition, this work can explore the use of Multi-Objective Evolutionary Algorithms (MOEAs) when finding the best data transformation for multi-target prediction models [23,24].

With respect to the training time of prediction models, some works have explored the use of multithreading and GPUs to diminish the computational cost. For example, GPUs are employed to accelerate the training time of a 3D convolutional neural network when classifying MRI Migraine Medical Data [25]. In [13], the use of GPUs improved the speedup (close to $20\times$) in the image texture extraction and analysis from medical images. Other work [26] has proposed a GPU cluster-based MapReduce framework to perform a disease probability prediction. Finally, GPUs have also been considered for accelerating a real-time gastrointestinal diseases detection [27]. Due to the aforementioned works, it is desirable to consider GPUs for accelerating the training time of predictive models.

To summarize, the SOFIA methodology presented in this paper brings together all the important aspects highlighted in this section. An extension of the panoramic prediction approach will be considered. This improvement will imply the extension of the data transformation proposed in [9] for supporting the optimization of every accuracy in multi-target prediction models instead of only optimizing their average. In addition, an extension of the NO metaheuristic to support MONO

will be presented. That implementation of MONO will allow parallel execution for diminishing the computational cost. Furthermore, the performance of MONO will be compared with some parallel MOEAs when finding the optimal data transformation. Finally, the proposed methodology is put into practice in the study of migraine, as the associated treatment requires several sessions to improve the quality of life of patients, diminishing the cost of collecting all medical features when selecting those medical features that need to be collected by doctors. That selected clinical features will allow us to know in advance whether the treatment will be effective in patients.

## 3. Methodology

### 3.1. The Sofia Method

The Selection Of medical Features by Induced Alterations in numeric labels (SOFIA) was designed for finding the best data transformation of the numeric labels while selecting the relevant features of clinical datasets from multi-target prediction models. This proposed technique improves any selected classification evaluation metric, such as the prediction accuracy or the F-score of every class attribute, without adding more columns to the dataset.

The selection of medical features was done when examining the list of optimal feature weights that allow us to obtain high values of the selected classification evaluation metric. The purpose of the SOFIA method is to find those weights that improve the representation of the numeric labels for each stage (feature weighting). This problem is multi-objective because we need to find the optimal weights that improve the selected classification metric for all stages in multi-target prediction models. In this sense, the relevant features will obtain high weighting values, whereas irrelevant features will have weighting values close to zero. Feature weighting can be used not only to improve any classification metric, but also to discard features with weights below a certain threshold $TH$ value and thereby increase the resource efficiency of the multi-target classifier.

With the purpose of performing small perturbations in numeric labels, this technique employs a multiplication and a rounding operation for each column of the medical dataset. In this sense, the objective of the multi-objective optimization metaheuristics is to achieve a set of efficient solutions, not dominated or Pareto front [28]. As a consequence of having more solutions, its computational cost was greater than algorithms with a single solution approach, specially when performing without parallelism [17]. For this reason, the proposed SOFIA method takes into consideration only those multi-objective optimization metaheuristics that allow a parallel execution, distributing the computational load on different cores of the computer and making an efficient execution [29].

The inputs of the SOFIA method are:

- An initial dataset $O$ containing $m$ clinical records, each containing the same set of $n$ features (columns) $c_1, c_2, c_3, \ldots, c_n$.
- The selected metaheuristic algorithm ($MA$) for performing the multi-objective optimization.
- The number $K$ of iterations to be performed by $MA$.
- A threshold $TH$ to discard features with weights below this value.
- The selected multi-target classification algorithm ($CA$) to build the prediction model $M$.
- The fitness function $F$, which refers to the selected evaluation metric (like F-score or accuracy) for measuring the precision of every $s$ stage in $M$.

The different steps in SOFIA are shown in Figure 1 and explained in the following lines:

1. The conversion of nominal labels of $O$ to numbers was done following a consecutive order of integers beginning with 1. It is done for the $n$ columns of $O$. The modified dataset was called $O'$ with $c'_1, c'_2, c'_3 \ldots, c'_n$ as modified columns.
2. Once the $O'$ dataset was generated, the next step was performing the feature weighting task. For it, the $MA$ found the optimal weights $w_j$, $1 \leq j \leq n$, i.e., one for each column $c'_j \in O'$ that

optimize the fitness values of every $s$ stage in multi-target prediction models. The $w_j$ weights will reflect the degree of relevance of a column $c'_j$ for a problem to solve, where $w_j \in \mathbb{R}\{0, 1\}$. Those $c'_j$ whose $w_j$ were lower than $TH$ were discarded. The current weighting vector solution $W_{sol}$ were shaped by those $w_j$ weights.

3.  The numeric labels of every cell $o'_{i,j} \in O'$ were multiplied by the corresponding weight $w_j$ through the $o'_{i,j} \times w_j$ operation, $\forall i, 1 \le i \le m$ and $\forall j, 1 \le j \le n$. This multiplication is illustrated in Figure 2.

4.  The $O'$ dataset was rounded to the tenth, generating the $O''$ modified dataset. The rounding to the tenth operation has been selected due the good results achieved in [9] when generating small perturbations among the different numeric labels in each column [30,31]. These rounded labels generated modifications in the prediction models learned by the multi-target classification algorithms that work with real numbers. An example of this step is shown in Figure 2.

5.  The prediction models were learned by the multi-target classification algorithm when it is trained with the modified dataset $O''$. The fitness value for each of the $s$ stages ($F_i^r$, $1 \le i \le s$) of $O''$ is obtained when performing the fitness function $F$ on it.

6.  The goals to be optimized by MA are be the maximization or minimization of the fitness value of each of the $s$ stages separately. If the current $W_{sol}$ is a non-dominated solution, the current Pareto front ($C_{pf}$) is updated when adding $W_{sol}$ to it, removing the dominated elements from the list.

The outputs of the SOFIA method was the list of optimal weighting vectors for performing the data transformation in the initial $O$ dataset when training the multi-target prediction model with the selected $CA$. These weighting vectors are collected in the $C_{pf}$ list.

It is important to mention that some steps of the SOFIA method were carried out with the help of metaheuristics. These steps are the second and the sixth. These, together with their intermediate steps were executed $K$ times, where $K$ is the number of iterations passed as parameter in the first step.

In order to perform a multi-objective optimization such as the accuracy or the F-score for all stages of multi-target classification models, some MOEAs were considered. In particular, those MOEAs that allow parallelism in their execution will be considered for reducing the computational cost in the optimization task [17]. In addition, some promising accuracy results have been achieved when performing a single-objective optimization with SA in a previous work [9]. For this reason, we believe in the convenience of extending the SA method to the multi-objective scenario.

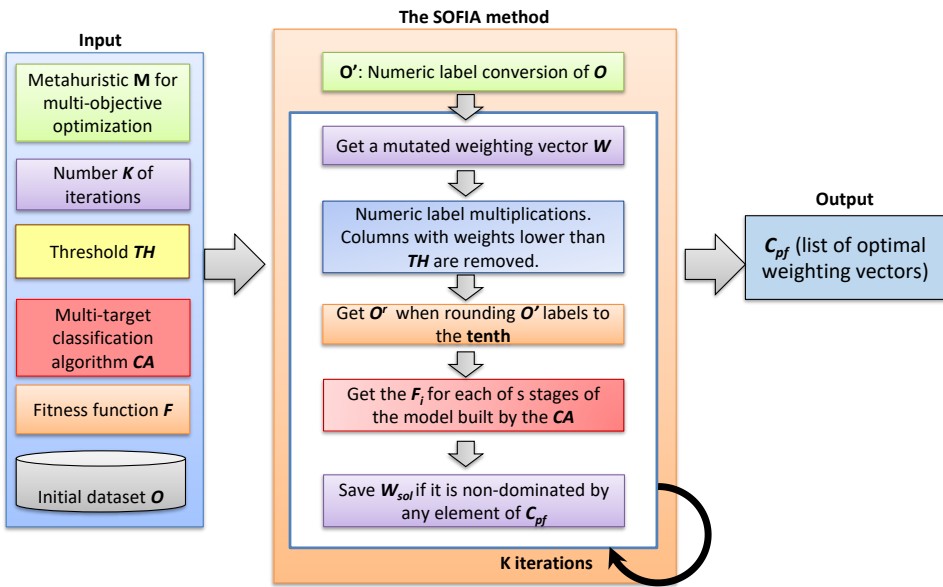

**Figure 1.** Diagram of the Selection Of medical Features by Induced Alterations in numeric labels (SOFIA) method.

| ***W*** (weights): | $w_1$ | $w_2$ | $w_3$ | | |
|---|---|---|---|---|---|
| | 0.795269560373731 | 0.18469775 | 0.767716221 | | |

| ***O'*** (dataset): | $c'_1$ | $c'_2$ | $c'_3$ | $y_1$ | $y_2$ |
|---|---|---|---|---|---|
| | 1 | 2 | 3 | low | high |
| | 2 | 3 | 1 | low | low |
| | 3 | 1 | 2 | high | high |

| $o'_{i,j}$ **x** $w_j$: | $c'_1$ **x** $w_1$ | $c'_2$ **x** $w_2$ | $c'_3$ **x** $w_3$ | $y_1$ | $y_2$ |
|---|---|---|---|---|---|
| | 0.79526956 | 0.369395501 | 2.303148663 | low | high |
| | 1.590539121 | 0.554093251 | 0.767716221 | low | low |
| | 2.385808681 | 0.18469775 | 1.535432442 | high | high |

| ***O''*** dataset: | $c''_1$ | $c''_2$ | $c''_3$ | $y_1$ | $y_2$ |
|---|---|---|---|---|---|
| | 0.8 | 0.4 | 2.3 | low | high |
| | 1.6 | 0.6 | 0.8 | low | low |
| | 2.4 | 0.2 | 1.5 | high | high |

**Figure 2.** Weighting dataset and rounding to the tenth.

### 3.2. Mono: Multi-Objective Natural Optimization Approach for Simulated Annealing

Due to the promising results achieved by [9,11] with the use of a single-objective NO [22], the present work has the purpose of extending it to support a multi-objective optimization.

In NO, the temperature does not need to be given because it is continuously tuned while running the SA algorithm. His proposed temperature metric is presented in Equation (1).

$$T = \frac{K \times (C_{min} - C_{init})}{N} \, , \tag{1}$$

where $N$ is the number of iterations, $K$ is a constant that refers to the backward degree and time/quality trade-off and has been set to 1, and $C_{min}$ and $C_{init}$ refer to the current minimal cost and initial cost, respectively. The energy difference is defined in Equation (2).

$$E_{diff} = C_{sol} - C_{min} \, , \tag{2}$$

where $C_{sol}$ is the cost of the solution. Finally, the probability ($P$) to compare with the random number ($R$) is given by Equation (3). $P$ is the probability of changing to a new solution. This is calculated when the accuracy is not lower than the fitness value. When $R \leq P$, SA moves the solution to another point within the search space to avoid being trapped in a local minimum.

$$P = e^{(-E_{diff}/T)} \, . \tag{3}$$

When extending the described NO method to the multi-objective scenario, it is important to note that the computation of the fitness value achieved by any solution is not trivial as in the single-objective optimization scenario. In fact, some metrics that consider all the optimization objectives have been proposed [32]. From them, the hypervolume metric was applied as a guidance criterion for measuring the fitness value of solutions in multi-objective optimization algorithms [33]. However, the computational time taken for computing that metric is a crucial factor for the performance of multi-objective optimization algorithms. In this respect, the Quick HyperVolume algorithm (QHV) [34] has obtained very competitive results in comparison with existing exact hypervolume algorithms, adding its advantage of computing the hypervolume metric with a low computational cost. Thus, QHV can be integrated with the proposed multi-objective NO (MONO) for the computation of the hypervolume metric. In this way, the criteria for accepting a new solution in NO will be modified for the multi-objective optimization scenario, taking into account those solutions that improve the

hypervolume metric instead of accepting those that improve a single-objective or the average of all the objectives.

All in all, the NO approach will be modified in this work for supporting multi-objective optimization and parallel execution. These changes are:

- At the beginning of the execution, multiple initial solutions ($C_{sol}$) were generated, one for each thread, instead of only handling one initial solution.
- $C_{sol}$ were saved in the Pareto front list ($C_{pf}$) when it is non-dominated by any element of $C_{pf}$ instead of replacing the current best solution ($C_{min}$) by $C_{sol}$ when its fitness value is lower than the obtained by $C_{min}$.
- Some changes in the computation of $E_{diff}$ and $T$ were done for extending Equation (3) to the multi-objective optimization. More specifically, the QHV method was employed to compute the hypervolume occupied by $C_{sol}$, $C_{pf}$ and $C_{init}$ for the multi-objective optimization. These changes are exposed in Equations (4) and (5).

$$E_{diff} = QHV(C_{sol}) - QHV(C_{pf}), \tag{4}$$

$$T = \frac{K \times (QHV(C_{pf}) - QHV(C_{init}))}{N}, \tag{5}$$

where $N$ is the number of iterations, $K$ is a constant that refers to the backward degree and time/quality trade-off and has been set to 1, and $C_{sol}$, $C_{pf}$ and $C_{init}$ refer to the current solution, Pareto front and initial solution, respectively.

In the next lines, the steps of the proposed Multi-Objective Natural Optimization (MONO) approach for SA are presented in detail:

1. The inputs of the algorithm are the number of iterations ($N$), the number of threads to perform ($Nth$) and the multi-objective fitness function.
2. A unique list of current non-dominated solutions ($C_{pf}$) is created. This list was accessible for all threads for allowing them to know whether a new solution is non-dominated by any of that list.
3. Different initial solutions ($C_{ini}$) were generated, one for each thread. Moreover, a number ($N$) of iterations to be performed for each thread was assigned.
4. For each thread:

    - In the beginning, the current solution ($C_{sol}$) was equal to the initial solution ($C_{ini}$).
    - A rollback solution ($C_{rb}$) was defined as the $C_{sol}$ before mutation. After that, the $C_{sol}$ was mutated.
    - The fitness value of the current solution ($C_{sol}$) was computed when performing the multi-objective fitness function.
    - A lock was applied for providing exclusive access to the $C_{pf}$ while the non-dominated comparison of any thread was performed.
    - The goodness of $C_{sol}$ was evaluated when verifying that it was not dominated by any of the elements of $C_{pf}$.
    - If the $C_{pf}$ list was empty or if $C_{sol}$ was non-dominated, $C_{sol}$ was added to the $C_{pf}$ list. After that, the $C_{pf}$ list was evaluated in order to remove any dominated solution.
    - The lock was removed.
    - If $C_{sol}$ was dominated by any element of the $C_{pf}$ list and for avoiding to get trapped in a local minimum, the R≤P comparison was evaluated, where R is a generated random number and P is computed with Equation (3) but applying the changes expressed in Equations (4) and (5). If R≤P is true, $C_{sol}$ is kept. Otherwise, a rollback is done when replacing $C_{sol}$ by $C_{rb}$.

5.  The task was performed on each thread until the assigned number of iterations was completed. After that, the non-dominated solutions are contained in the $C_{pf}$ list.

Regarding the MONO implementation on the GPU, the CPU will carry out the next operations: (1) initializing the MONO parameters, (2) finding a random initial solution, (3) assigning the MONO iterations to be performed for each thread, and (4) presenting the global Pareto solution list. At the beginning, all threads had the same initial solution as the current local solution. On the other hand, the GPU carried out the next thread operations: (1) obtaining the fitness value for the current local solution, (2) comparing and updating the global Pareto solution list, and (3) changing the current local solution according to the MONO algorithm.

## 4. Analysis Case

In this section, we will apply the SOFIA method in a real case involving a chronic disease, namely the improvement of multi-target prediction models that classify every treatment response when treating migraines with OnabotulinumtoxinA (BoNT-A). This case study was selected due to the complexity of the problem in terms of modeling and the selection of variables, as well as its socio-economic impact. As has been mentioned in [35], it would be very useful to know beforehand which patients will respond and which will not. Knowing the phenotype–response relationship may help in the development of new treatments for the 20–30% of patients that do not respond to the treatment [36]. Besides the cost, it would prevent the patients from suffering the pain associated with the treatment.

The issues involved in the use of the SOFIA method when improving the multi-target prediction models for the BoNT-A treatment will be described in this section. Figure 3 presents the experiment workflow on which this experiment was based. Firstly, a database was loaded with the medical records from the two participating hospitals. Secondly, the class attribute was defined by considering the retrospective data available. Thirdly, clinical features were categorized in order to work with homogeneous data. After that, some parallel multi-objective metaheuristics are employed to perform the SOFIA method. In addition, a multi-target classification algorithm like predictive clustering trees (PCT) is applied for training multi-target prediction models.

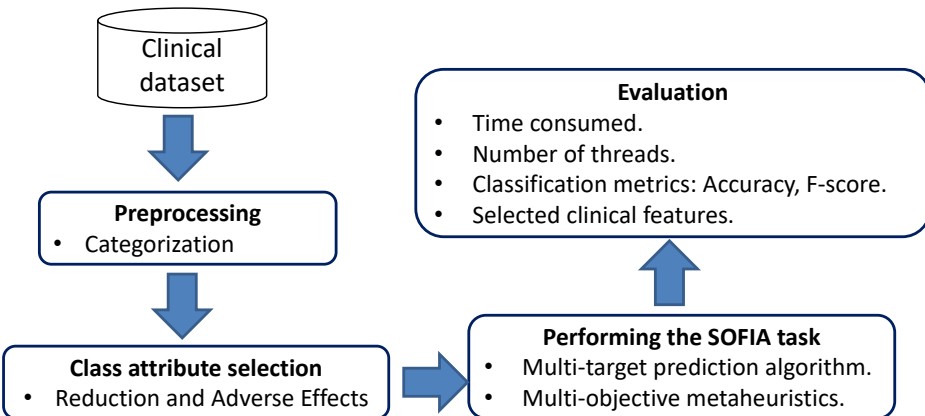

**Figure 3.** SOFIA experiment workflow.

### 4.1. Clinical Dataset

Retrospective medical data have been collected from various medical histories of patients under previous or current chronic migraine treatment with BoNT-A with follow-up at the Headache unit of two tertiary-level hospitals.

The number of patients collected from two hospitals were 173 (116 from Hospital Clínico Universitario in Valladolid and 57 from Hospital Universitario de La Princesa, in Madrid). A total

of sixty-two baseline features have been categorized. Collected medical features were related to the following points: clinical pain features, demographic features of patients, comorbidities, tested and concomitant preventive drugs, pain impact measures, and available analytical parameters. The latter were obtained from blood tests recorded in the clinical history which were performed for other reasons in the three months prior to, or three months after, the first infiltration, and included hemogram and liver, renal, thyroid, ferric, vitamin B12, folic acid and vitamin D profiles. The efficacy of BoNT-A was evaluated by comparing the baseline situation (before the first infiltration) and the situation after 12–16 weeks following each of the infiltrations, through the following parameters: number of days of pain per month, percentage reduction in days with pain, subjective intensity of pain, number of days of disability due to pain per month, drug consumption for pain and adverse effects of infiltration. Since this was a retrospective study, not all the data could be obtained for each patient in a systematic way.

In order to train the treatment response prediction models, clinical data need to be previously processed in order to achieve a high level of accuracy. In fact, some patients are non-respondent, while others respond after the *i*th session. With the purpose of predicting the outcome of all sessions, multi-target prediction models are considered. Nevertheless, problems like having a small set of patients with many features, still need to be solved. The incompleteness of data, continuous numeric values, and medically categorized data are difficulties to tackle in our medical dataset. All in all, it is hard to properly process all this information for training prediction models with high precision through to classification algorithms. Table 1 presents example of continuous and categorized data available in our clinical dataset.

**Table 1.** Example of continuous and categorized features available in the clinical dataset.

| Toxin-Age of Onset (Years) | Body Mass Index (kg/m$^2$) | Hemoglobin (g/dL) | Creatinine (mg/dL) | Platelets (u/mcL) | Reduction Effects (1–4) |
|---|---|---|---|---|---|
| 51 | 20.39 | 13.4 | 0.71 | 213,000 | 4 |
| 49 | 26.5 | 14.2 | 0.55 | 252,000 | 2 |
| 36 | 23.15 | 13.5 | 0.44 | 304,000 | 3 |
| 26 | 17.7 | 13.1 | 0.66 | 218,000 | 2 |
| 31 | NA | 14.8 | 0.71 | 327,000 | 1 |
| 50 | NA | 16.2 | 0.74 | 327,000 | 3 |

## 4.2. Categorization of Clinical Features

To solve the heterogeneity in the collected medical datasets, the values are categorized. The method selected for performing the categorization of the medical data is based on the mean and standard deviation. Applying this method is possible when working with more homogeneous values.

The mean and standard deviation categorization type centers the intervals around the mean ($\mu$), and defines subsequent intervals by adding or subtracting the standard deviation ($\sigma$). For instance, three categories are defined for each of our clinical features. The intervals $(V_{min}, \mu - \sigma)$, $(\mu - \sigma, \mu + \sigma)$ and $(\mu + \sigma, V_{max})$ are used to refer to value 1, value 2 and value 3, respectively. It should be noted that $V_{min}$ and $V_{max}$ are the minimum and maximum values of the data, respectively.

## 4.3. Class Attribute Selection

With the purpose of measuring how efficient a treatment stage has been, it is necessary to define the class attribute. This refers to the clinical feature used to measure the effectiveness of treatment. For the migraine treatment, HIT6 value [37], intensity, duration and frequency of attacks [38] are good candidates for class attributes according to doctors. However, the values of some of these features are not usually provided in our medical dataset. In fact, the HIT6 value is missing in many of our clinical records. In consequence, a combination of the reduction (R) and the adverse (A) effects, which are those features more frequently found in our database, has been selected to define the class attribute. Reduction and adverse effects are defined with values directly provided by doctors.

These clinical features, *R* and *A*, are measurable values from an objective point of view based on definitions. *R* takes values from 1 to 4 according to the percentage of reduction of days of migraines. It takes the value of 1 when the percentage reduction of days of migraine is less than or equal to 25%, 2 for the interval between 25% and 49%, 3 for the interval between 50% and 74% and 4 when the percentage is greater than or equal to 75%. *A* is equal to 1 when there are no adverse effects, 2 when there are mild adverse effects (easily tolerated), 3 when there are moderate adverse effects (interfere with usual activities and may require suspension of treatment) and 4 when there are serious adverse effects (incapacitate or disable usual activities, and require suspension of treatment as well as medical intervention) [2].

High levels of *R* indicate good treatment results, while high levels of *A* point to many adverse effects. With the purpose of obtaining a directly proportional feature, our class attribute ($N_{AC}$) has been determined by dividing *R* and *A*.

In this work, a similar approach as the one based on HIT6 [37] (two response categories: low and high [36] has been considered for class attribute categorization, instead of the three categories (low, medium and high) used for the rest of the clinical features. Lower responses are labeled when the $N_{AC}$ value falls into the ($V_{min}$, cut-off point) interval, while high response labels are used for those values falling within the (cut-off point, $V_{max}$) interval. In this case, $V_{min} = 0.25$ occurs when $R = 1$ and $A = 4$, while $V_{max} = 4$ occurs when $R = 4$ and $A = 1$. We select a cut-off point of 1.40. The reason to use this value is the fact of trying to emulate the criterion used of the 30% decrease in the HIT6 value. The treatment is considered to be effective if the HIT6 is reduced in a 30% or more, according to the PREEMPT clinical trial [36]. In this way, values lower than 1.40 represent the 30% of the values that $N_{AC}$ can take. Then, the low and high categories are defined with the intervals (0.25, 1.40) and (1.40, 4), respectively. Table 2 depicts an instance of the $N_{AC}$ computation using different values provided by the hospitals.

**Table 2.** Example of the class attribute categorization.

| Reduction Effects (R) | Adverse Effects (A) | R/A | Categorized Value |
|:---:|:---:|:---:|:---:|
| 1 | 1 | 1 | low |
| 2 | 1 | 2 | high |
| 3 | 2 | 1.5 | high |
| 1 | 2 | 0.5 | low |

Applying the class attribute defined here for both stages of the treatment, the distribution of high–low values was obtained over the records of the medical dataset, as shown in Table 3. This results in the following baseline values of accuracy: 56.64% and 58.95% for the first and second stage, respectively. These values refer to classifying all the records with the most frequent label for each stage of the treatment. More in detail, 98 and 71 have a high response for the first and second stages of the treatment. On the other hand, 75 and 102 patients have a low response for the first and second stages of the treatment.

**Table 3.** Distribution of high-low categories through both stages.

| Response | Stage 1 | Stage 2 |
|:---:|:---:|:---:|
| high | 98 | 71 |
| low | 75 | 102 |

### 4.4. Performing the Sofia Method

For performing the SOFIA method, it is important to define its input parameters. Regarding the multi-target classification algorithm (CA) to use the predictive clustering trees combined with the Random tree (PCT + RT) method was considered due the promising results achieved in [9]. In that

study, several classification algorithms were tested when improving their prediction models with the feature weighting task. The NO metaheuristic was the method employed for performing the optimization. In their results, the PCT + RT was the best combination with an improved accuracy close to 85% when predicting the response to the first and second stages of treatment. Random tree (RT) is a non-deterministic algorithm that builds a tree considering K randomly chosen features for each node. Authors conclude that, as a result of being a non-deterministic algorithm, RT was benefited with the use of NO. In contrast to the deterministic algorithms, it allowed a deeper exploration of the search space to avoid being trapped in a local minimum.

Furthermore, we will consider some metaheuristic algorithms (MA) that allow a parallel execution for reducing the computational cost in the optimization task [17]. Algorithms to be considered were: GDE3 [39], PESA2 [40], SMPSO [41], NSGA-II [42], NSGA-III [43] and SPEA2 [44]. All the aforementioned methods were implemented in the MOEA framework presented at [45]. In addition, our proposed MONO metaheuristic was considered.

The threshold $TH$ has been set to 0.10 because below this value, imperceptible changes are made in the data transformation proposed in the SOFIA method (multiplication and rounding operations). As previously mentioned, those features whose weights are less than $TH$ were discarded when training the multi-target prediction models.

According to [46,47], the classification model evaluation metrics typically used in papers are: accuracy, sensitivity, specificity and F-score. From them, Demvsar [46] mentions that classification algorithms are mainly compared by accuracy. In fact, it can be used when the class distribution is similar, while the F-score is a better metric when there are imbalanced classes [47]. It is true that the accuracy metric does not distinguish between the number of correct labels of different classes. For this reason, we will present the sensitivity, specificity and F-score values achieved by prediction models, improved with the SOFIA method.

## 5. Experiments

In this section, a comparison of runtime, accuracy and selected features between the proposed MONO metaheuristic and some MOEAs is presented. Those methods are employed to improve the accuracy of multi-target prediction models through the SOFIA method. Prediction models are learned with the aforementioned PCT + RT classification method [9].

To compare the results of this experiment with our previous work [9], we selected the accuracy metric for every stage of the BoNT-A treatment as the goal to maximize ($F$). In this experiment, we contemplated two stages, so there were two objectives to be maximized simultaneously, i.e., $F_1$ and $F_2$. Positive and negative values refer to "high" and "low" responses, respectively. In addition, sensitivity and specificity refer to the proportion of true positives and true negatives that are correctly identified as such, respectively.

In the experiments, the $K$ number of iterations was established in $10^6$ as in [9]. The population size for MOEAs was established in 100. This value was selected in order to guarantee the diversity of solutions while avoiding a slow convergence of individuals [48,49].

### 5.1. Runtime

The number of CPU threads that were considered in this experiment were: 1 (no parallelism), 2, 4, 6, 8, 10, 12, 14, 16, 18, 20 and 22. The server used to perform the experiments has four Intel Xeon E7-4830 CPUs 2.13GHz, each one with eight cores and 16 threads (32 cores and 64 threads overall) and 256 GB of RAM. The methods performed in this section have been implemented in Java. Moreover, MONO and the other methods have also been run on a GPU with the help of TornadoVM [50]. The NVIDIA GeForce GTX Titan X (3072 cores, 1.0 GHz) has been employed to perform the GPU experiments.

Table 4 presents the execution time of the previously employed algorithms following the hour:minutes:seconds format. The methods was performed on CPU and GPU three times to obtain the average runtime.

**Table 4.** Time performance of Multi-Objective Natural Optimization (MONO), other parallel Simulated Annealing (SA) approaches and multiobjective evolutionary algorithms (MOEAs) metaheuristics on the SOFIA method.

| Methods | Number of Threads | | | | | | | | | | | | |
|---|---|---|---|---|---|---|---|---|---|---|---|---|---|
| | 1 | 2 | 4 | 6 | 8 | 10 | 12 | 14 | 16 | 18 | 20 | 22 | GPU |
| MONO | 8:10:14 | 4:07:31 | 2:29:23 | 2:05:51 | 1:34:38 | 1:13:27 | 1:07:41 | 1:01:53 | 0:56:47 | 0:52:15 | 0:49:51 | 0:49:48 | 0:44:31 |
| PSA | 8:15:28 | 4:05:19 | 2:28:43 | 2:05:27 | 1:34:21 | 1:12:15 | 1:06:34 | 1:00:21 | 0:55:13 | 0:51:30 | 0:48:42 | 0:49:11 | 0:44:23 |
| SPEA2 | 8:16:53 | 4:06:08 | 2:30:37 | 2:04:26 | 1:33:47 | 1:11:02 | 1:06:58 | 1:01:39 | 0:54:48 | 0:51:09 | 0:49:14 | 0:49:32 | 0:45:05 |
| NSGAIII | 8:12:20 | 4:05:14 | 2:28:52 | 2:03:54 | 1:34:02 | 1:12:31 | 1:07:42 | 1:00:45 | 0:55:34 | 0:51:48 | 0:48:57 | 0:49:05 | 0:44:48 |
| NSGAII | 8:12:04 | 4:05:12 | 2:28:48 | 2:04:35 | 1:33:10 | 1:10:46 | 1:06:35 | 1:01:26 | 0:55:03 | 0:51:20 | 0:49:35 | 0:49:21 | 0:44:26 |
| SMPSO | 8:13:02 | 4:06:16 | 2:29:04 | 2:06:17 | 1:33:51 | 1:13:15 | 1:07:06 | 1:00:31 | 0:56:19 | 0:52:04 | 0:49:13 | 0:49:34 | 0:44:14 |
| PESA2 | 8:16:53 | 4:07:01 | 2:29:27 | 2:05:21 | 1:34:25 | 1:12:09 | 1:07:29 | 1:01:15 | 0:55:28 | 0:51:33 | 0:48:35 | 0:48:49 | 0:44:57 |
| GDE3 | 8:13:02 | 4:06:08 | 2:29:45 | 2:04:38 | 1:33:29 | 1:10:38 | 1:06:24 | 1:00:55 | 0:54:35 | 0:51:17 | 0:49:25 | 0:49:15 | 0:45:38 |

According to the results, the employed parallel optimization metaheuristics benefited from the use of more threads to distribute the computational load in the SOFIA method. It can be observed that all the metaheuristic methods perform with similar runtime when applying the same configuration of threads. On the other hand, the time difference between 20 and 22 threads was much smaller than the difference between the other thread configurations. In fact, we tested with more threads but we noticed that results were similar. Moreover, time was close to 45 minutes when running the methods on the GPU. All in all, with the use of GPU, the runtime was reduced more than 10.5 times (from more than 8 h to less than 45 min) approximately.

*5.2. Accuracy*

Table 5 presents the best accuracy values when improving multi-target models through the SOFIA method for the first and second stages of the BoNT-A treatment, with the corresponding hardware (HW) configuration that allows us to achieve these results. To present the results of this table, we selected, for each multi-objective metaheuristic, those non-dominated solutions that have the highest accuracies $F_1$ and $F_2$ (fitness value of the first and second stages, respectively) as described in Section 4.4. Moreover, we have added some comparison results. The first method presented in Table 5 refers to results achieved when training prediction models only with the PCT+RT classifier algorithm (without any optimization). In the row below, the NO [22] was applied to optimize accuracies in prediction models trained with RT through the SAR encoding method [9]. Since the NO method does not perform a multi-objective optimization, the accuracy optimization of single-target prediction models (trained with RT) has involved two operations, one for each stage of the BoNT-A treatment, as performed in [11]. For the next rows of Table 5, MONO and MOEAs methods are employed to perform the SOFIA optimization. The results are as follows:

**Table 5.** Performance metrics of parallel MOEAs and MONO in combination with the predictive clustering trees and random tree (PCT + RT) multi-target classification algorithm for performing the SOFIA method. Best accuracies are highlighted in bold.

| Metaheuristic Algorithm | HW | First Stage | | | | Second Stage | | | |
|---|---|---|---|---|---|---|---|---|---|
| | | Accuracy | Sensitivity | Specificity | F-Score | Accuracy | Sensitivity | Specificity | F-Score |
| No metaheuristic | 1 thread | 61.63% | 65.17% | 25.84% | 63,95% | 62.79% | 75.14% | 57.61% | 60.45% |
| FSS + RT (without SAR) | 1 thread | 68.14% | 64.78% | 70.25% | 66.13% | 67.45% | 70.21% | 65.18% | 65.43% |
| NO + RT [11] | 1 thread | 84.93% | 87.56% | 81.45% | 82.35% | 85.74% | 83.24% | 88.14% | 83.17% |
| MONO | 8 threads | **88.62%** | 89.58% | 86.58% | 87.91% | **87.42%** | 85.16% | 89.61% | 86.73% |
| PSA | GPU | 83.67% | 84.17% | 80.38% | 80.07% | 85.23% | 83.49% | 89.61% | 85.18% |
| SPEA2 | GPU | 79.48% | 75.94% | 81.45% | 77.15% | 76.19% | 81.24% | 75.56% | 77.18% |
| NSGAIII | 16 threads | 81.39% | 77.61% | 83.29% | 79.17% | 84.88% | 83.29% | 78.14% | 80.52% |
| NSGAII | 8 threads | 84.17% | 83.29% | 78.71% | 82.28% | 85.96% | 86.26% | 81.95% | 83.62% |
| SMPSO | GPU | 79.06% | 82.31% | 75.09% | 78.35% | 82.56% | 81.45% | 84.26% | 80.94% |
| PESA2 | 8 threads | 76.74% | 82.31% | 71.25% | 73.56% | 84.88% | 76.56% | 82.31% | 74.21% |
| GDE3 | 18 threads | 82.56% | 85.74% | 77.31% | 77.52% | 81.39% | 76.29% | 83.29% | 77.08% |

According to the results, high values of accuracy, sensitivity, specificity and F-score are obtained when the multi-target prediction models were improved with the SOFIA method. Table 5 presents, in every method, similar values of accuracy and F-score. For example, SPEA2 had 79.48% and 77.15% values of accuracy and F-score for the first stage of the BoNT-A treatment when using the GPU. This situation is the consequence of having almost balanced data [47], with percentages close to 56% and 41% for "high" responses, and 43% and 58% for "low" responses in the first and second stages respectively.

Regarding the HW configuration, PSA, SPEA2 and SMPSO achieved the best trade-off in accuracy when performing on the GPU. Furthermore, MONO, NSGAII and PESA2 achieved the best results when performing on eight threads. Finally, NSGAIII and GDE3 have needed 16 and 18 threads to achieve high results respectively.

In terms of accuracy, the NO+RT combination [11] achieved high percentages in every evaluation metric considered. However, it was not able to maximize the accuracy for both stages at the same time, while the MOEAs and MONO methods were able to do this. In fact, the MONO metaheuristic achieved the best accuracy in both stages (88.62% and 87.42%, respectively) when using eight threads. Moreover, sensitivity and specificity values were higher than 85%, indicating a low number of false positives and false negatives. As in any parallel SA approach [51] with the use of more threads, the MONO method explored the solution space more extensively, achieving the best accuracies in our experiment.

### 5.3. Trade-Off Study

With the purpose of studying the trade-offs, Figure 4 is presented. This figure shows the runtime and the accuracy obtained by the methods contained in Table 5. Figure 4a,b depict the accuracies and runtimes produced during the SOFIA method for the first and second stages of the BoNT-A treatment, respectively. In the figures, the best points in terms of accuracy are marked with red circles. It is important to note how the charts show a better runtime performance for metaheuristic methods when using GPU while achieving high accuracy percentages. Moreover, in the MONO method, the GPU result achieved high accuracies in less than 45 min. All in all, with the GPU, a good exploration of the solution space has been achieved, being close to the accuracy obtained with eight threads.

Given that it is difficult for us to visualize which method maximizes the accuracy (i.e., minimize the error) for both stages simultaneously, Figure 5 is presented considering only multi-objetcive optimization metaheuristics. In Figure 4a,b, MONO achieved a high accuracy and a low computational cost in every stage when running on the GPU. However, the best accuracy percentage is achieved when performing MONO on eight threads for every stage. In fact, in Figure 5, the best tradeoff was achieved with MONO when performing on eight threads (marked with a red circle), achieving 88.32% and 87.42% accuracy for the first and second stages respectively.

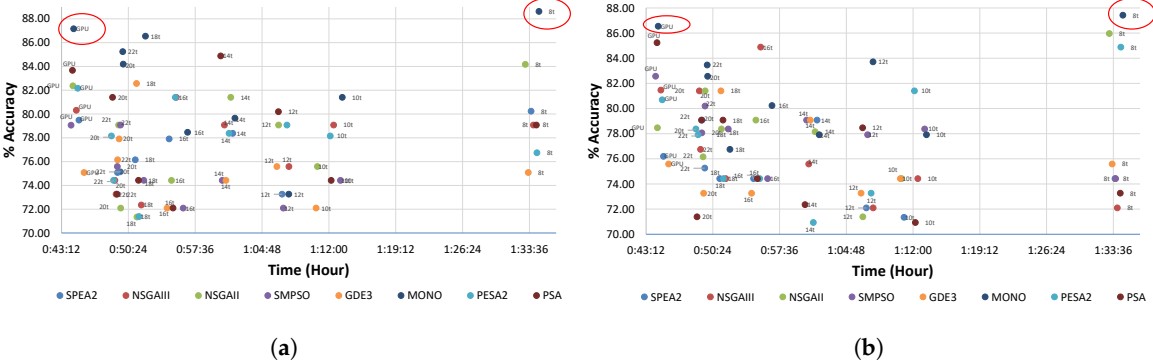

**Figure 4.** Time vs accuracy in the first and second stages. (**a**) first stage; (**b**) second stage.

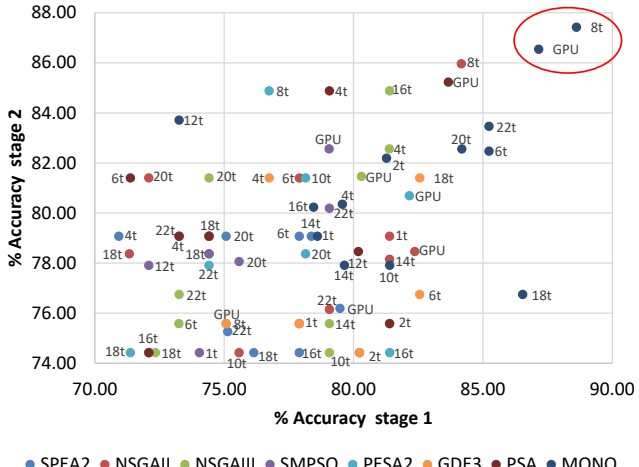

**Figure 5.** Best points for each thread setting and metaheuristic method.

### 5.4. Selection of Medical Features

With the purpose of understanding the pathophysiological features that determine the "high" or "low" response to the BoNT-A treatment, we examined the multi-target prediction models for finding those explicit features that appears in them. In order to retain only relevant features, the aforementioned threshold $T = 0.10$ was applied when performing the SOFIA method. Thus, the features that appears in prediction models of Table 6 are those whose weights were higher than $T$ when performing the SOFIA method.

**Table 6.** Selected clinical features by the SOFIA method when performing multi-target prediction models with MONO and parallel MOEAs.

| Metaheuristic Applied | Number of Features | Selected Clinical Features |
|---|---|---|
| FSS+RT (without SAR) | 11 | Sex, Chronic migraine time evolution, GON, Drugs tested before toxin, Preventive oral treatment, catamenial, Concomitant treatment with statins, Gastropathy, Pneumopathy, Headache days per month, Analgesics days per month |
| NO+RT [11] | 16 | Retroocular component, GGT, Migraine days per month, Drugs tested before toxin, Neuromodulator, Concomitant antihypertensive treatment, Enolism, Analgesics days per month, 1st grade family with migraine, Unilateral pain, GON, Chronic migraine time evolution, Anxiety, Platelets, Pneumopathy, Serum iron |
| MONO | 14 | Headache days per month, Migraine days per month, Chronic migraine time evolution, GON, Hemoglobin, Analgesic abuse, Serum iron, 1st grade family with migraine, Retroocular component, Unilateral pain, Platelets, Anxiety, Concomitant oral preventive treatment, Onset age of toxin treatment |
| SPEA2 | 14 | Chronic migraine time evolution, Hemoglobin, Analgesic abuse, Retroocular component, GON, Anxiety, Onset age of toxin treatment, 1st grade family with migraine, Headache days per month, GGT, Migraine days per month, Drugs tested before toxin, Neuromodulator, tricyclic antidepressants |
| NSGAIII | 15 | Onset age of toxin treatment, Migraine days per month, Chronic migraine time evolution, drugs tested before toxin, vitamin B12, GON, Preventive oral treatment at time of infiltration, Tricyclic antidepressants, Calcium antagonists, Catamenial, Concomitant oral preventive treatment, Gastropathy, Headache days per month, Analgesic abuse, Unilateral pain |
| NSGAII | 11 | Unilateral pain, GON, 1st grade family with migraine, Onset age of toxin treatment, Serum iron, Creatinine, Analgesic abuse, Preventive oral treatment at time of infiltration, Neuromodulator, Concomitant antidepressant treatment, Chronic migraine time evolution, Retroocular component |
| SMPSO | 11 | Headache days per month, GGT, Migraine days per month, Drugs tested before toxin, Neuromodulator, Concomitant oral preventive treatment, Enolism, Analgesic abuse, 1st grade family with migraine, Unilateral pain, GON |
| PESA2 | 10 | Chronic migraine time evolution, Anxiety, 1st grade family with migraine, Analgesic abuse, Platelets, Headache days per month, Unilateral pain, Migraine days per month, GON, Onset age of toxin treatment |
| GDE3 | 15 | Drugs tested before toxin, Concomitant oral preventive treatment, Headache days per month, Enolism, Analgesic abuse, 1st grade family with migraine, GGT, Migraine days per month, Chronic migraine time evolution, Hemoglobin, Retroocular component, GON, Anxiety, Onset age of toxin treatment, tricyclic antidepressants |

In general, predictive models presented in Table 6 need to know the clinical values of 10 to 15 features for predicting the response to the two stages of treatment with BoNT-A. With the FSS technique, 11 medical features are required to collect when performing the prediction, but it achieved low accuracies for both stages, as presented in Section 5.2. Considering the accuracy values obtained in Table 5, we prefer to analyze the features involved in the two prediction models that achieved the best accuracies with the SOFIA method, these were achieved when applying the NSGA II and MONO metaheuristics. It could be preferred to use the NSGA II optimized model since it only uses 11 features while MONO uses 14. However, this decision implies losing some accuracy, since NSGA II achieves accuracies close to 85% for both stages while MONO achieves accuracies close to 88%.

Table 7 presents the frequency of clinical features that were selected by the SOFIA method in the nine metaheuristic methods of Table 6. When analyzing their clinical features involved for predicting the treatment responses to BoNT-A, it can be observed that both models need to know from patients the next common clinical features: GON, Chronic migraine time evolution, Analgesic abuse, first-grade family with migraine, retroocular component, unilateral pain, serum iron, onset age of toxin treatment. Therefore, the convenience of choosing one model or another will depend on collecting the next additional clinical values: creatinine, preventive oral treatment at the time of infiltration, neuromodulator, concomitant antidepressant treatment, for the model optimized with NSGAII; or headache days per month, migraine days per month, hemoglobin, platelets, anxiety, concomitant oral preventive treatment, for the model optimized with MONO.

**Table 7.** Frequency of selected clinical features by the SOFIA method in the nine metaheuristic methods of Table 6.

| Clinical Features | Frequency |
|---|---|
| GON | 9 |
| Chronic migraine time evolution | 8 |
| Headache days per month | 7 |
| Migraine days per month | 7 |
| 1st grade family with migraine | 7 |
| Analgesic abuse | 7 |
| Drugs tested before toxin | 6 |
| Unilateral pain | 6 |
| Onset age of toxin treatment | 6 |
| Retroocular component | 5 |
| Anxiety | 5 |
| GGT | 4 |
| Neuromodulator | 4 |
| Preventive oral treatment | 3 |
| Enolism | 3 |
| Platelets | 3 |
| Serum iron | 3 |
| Hemoglobin | 3 |
| Tricyclic antidepressants | 3 |
| Catamenial | 2 |
| Gastropathy | 2 |
| Pneumopathy | 2 |
| Analgesics days per month | 2 |
| Sex | 1 |
| Concomitant treatment with statins | 1 |
| Concomitant antihypertensive treatment | 1 |

All in all, the SOFIA method has allowed us to optimize the accuracy of multi-target predictive models related to the BoNT-A treatment, reducing the number of clinical features to collect, from 62 to 15 or less, saving time in collecting medical information and generating a prediction of both stages close to 85% for the model using NSGAII and 88% for the model using MONO.

## 6. Conclusions

This work presents the improvements in multi-target prediction models through the SOFIA method. The proposed methodology makes use of a data transformation when converting the categorized labels into numbers, multiplying them by a feature vector, and rounding the results to the tenth. The feature weighting vector will reflect the relevance of every feature to the predictive model and it will be the solution to be found by multi-objective metaheuristic methods. For running the SOFIA method, it is necessary to define some inputs parameters such as: the goal to optimize (accuracy, F-score, etc.), the number of iterations, the threshold, the multi-target classifier algorithms and the metaheuristics selected to perform the optimization. The SOFIA method has decreased the number of medical features required to perform the treatment prediction, from 62 to less than or equal to 15 features, while achieving high accuracies in both stages of treatment.

Moreover, the MONO method, a multi-objective version of the NO metaheuristic, has been proposed. This method allows the use of highly parallel hardware, as many cores CPU or the GPU employed in the experiments.

Both proposed methods have been applied to perform the optimization on a treatment that involves multiple stages of treatment like the migraine. More specifically, we used the proposed methods to improve the accuracy for every stage when predicting the treatment response to two stages of the BoNT-A treatment. When performing the SOFIA method with the MONO metaheuristic on eight threads, accuracies close to 88% for both stages have been achieved, taking around one and a half hours to train the optimal model and outperforming the other metaheuristics in accuracy terms. Furthermore, when running on the GPU, close accuracies were achieved in less than 45 min. We can conclude that the SOFIA method and the MONO metaheuristic have provided an interesting combination to improve the accuracy of classification models for the migraine scenario.

**Author Contributions:** All the authors contributed equally to this work. F.P.B., A.A.D.B.G., L.M.S.R. and J.L.A. participated in the conceptualization, methodology, implementing and testing of the research. They also discussed the basic structure of the manuscript, drafted its main parts, and reviewed and edited the draft. All authors have read and agreed to the published version of the manuscript.

**Funding:** This research has been supported by the Spanish MICINN under project PID2019-110866RB-I00. The project was co-financed by the Ministry of Education, Science, Technology and Innovation (SENESCYT) of the Government of the Republic of Ecuador (8905-AR5G-2016) and the European Union's Horizon2020 Framework Programme for Research and Innovation under grant H2020-ICT-2017-779656 (HIPEAC collaboration grant-2019).

**Acknowledgments:** The authors also want to express their gratitude to the Service of Neurology of Hospital Universitario La Princesa in Madrid and Hospital Clínico Universitario in Valladolid, whose help has been precious for this work. In particular, Ana Gago, Mercedes Gallego, Marina Ruiz and Ángel Guerrero.

**Conflicts of Interest:** The authors declare no conflict of interest.

## Abbreviations

The following abbreviations are used in this manuscript:

| | |
|---|---|
| SOFIA | Selection Of medical Features by Induced Alterations |
| BoNT-A | OnabotulinumtoxinA |
| SA | Simulated annealing |
| PSA | Parallel Simulated annealing |
| RT | Random trees |
| PCT | Predictive clustering trees |
| NO | Natural optimization |
| MONO | Multi-objective Natural optimizationapproach for Simulated Annealing |
| MOEA | Multi-objective evolutionary algorithms |
| GPU | Graphics processing unit |
| HW | Hardware |
| FSS | Feature Subset Selection |
| EMRs | Electronic medical records |

GDE3    third version of the generalized differential evolution
PESA2   Pareto Envelope-based Selection Algorithm
SMPSO   Speed-constrained Multi-objective Particle Swarm Optimisation Algorithm
NSGA    Non-dominated Sorting Genetic Algorithm

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
