# Peer review of "SOFIA: Selection of Medical Features by Induced Alterations in Numeric Labels"

_electronics, doi:10.3390/electronics9091492_

Round 1

Reviewer 1 Report

This manuscript represent a really interesting approach of a health related problem through ML. And represent a sound topic. 

As a recommendation for authors, in future manuscript. The information need to be more concise, the manuscript is too large and is not designed for an overall public (For example, medicians interested in this field), this limit the potential target audience for such a good research. 

Again, I consider the manuscript needs to be published.

However, before, include abbreviations in tables.  

Congratulations. 

Author Response

This manuscript represents a really interesting approach of a health related
problem through ML. And represent a sound topic. 

As a recommendation for authors, in future manuscript. The information need to
be more concise, the manuscript is too large and is not designed for an overall
public (For example, medicians interested in this field), this limit the potential
target audience for such a good research. 

Again, I consider the manuscript needs to be published.

However, before, include abbreviations in tables.  

Thanks for the annotation. More abbreviations have been included in the
“abbreviations” section. In addition, some redundant parts of sections 1 and 2
have been removed to shorten the manuscript. More specifically, the next lines
have been removed in section 1:

“The use of EMRs makes it possible to efficiently manage the medical diagnosis
and health in patients, thus improving the patient care”

“Regarding the medical treatments collected in EMRs, some records refer to
continuous treatments, whose purpose is to mitigate or eliminate the patient’s
symptoms, for example, treatments for chronic diseases such as Parkinson [7]
or migraine [2]. Other EMRs refer to treatments for mitigating short-term
illnesses like the flu or cough [8]. Some treatments, specially continuous
treatments, are very expensive. In fact, people with chronic conditions typically
have more health needs at any age, so the associated costs become
disproportionately high [9]. For example, patients suffering from Parkinson’s
disease usually discontinue the treatment due to its ineffectiveness when
mitigating the pain [10], which thus involves wasting money. In order to avoid
this, cost-benefit analyses have been applied for patients with continuous
treatment, such as those that suffer hepatitis C [4,5] or migraine [6]. Due to the
economic cost of treatments, it is important for our methodology to be able to
know in advance whether a treatment will be effective in patients with continuous treatments. It can be achieved when predicting the response of all
stages of treatment at the same time (multi-target prediction).”

Moreover, the next lines have been removed in section 2:

“Computational models that allow the prediction to all responses of a multi-
stage treatment can leverage the use of electronic medical records (EMRs). In
fact, modern hospitals possess a wide variety of monitoring and data collection
devices that provide relatively inexpensive means of collecting and storing data
in inter and intrahospital information systems [23]. Given the rapid growth of
EMRs, the traditional analysis of data collected by medical experts need to be
combined with computational methods to help in the decision making process of
a specific continuous treatment. In this respect, the use of machine learning
techniques has made it possible to tackle the analysis of medical data and the
construction of prediction (classification) models [24,25].”

Reviewer 2 Report

Dear Authors,

I have read the manuscprit and I have a question for you:

- Methods: please clarify the methods used to choose the references.

Author Response

Dear Authors,
I have read the manuscript and I have a question for you:
- Methods: please clarify the methods used to choose the references.

Methods used for comparing with the proposed MONO metaheuristic were
selected because they are state-of-the-art algorithms to perform a multi-
objective optimization in a parallel way. The PSA pethood is a well-known
Simulated Annealing method that performs in parallel. Methods like GDE3,
PESA2, NSGAII, SMPSO and others, are widely employed when solving multi-
objective optimization problems. Because of that, we have considered to
compare them with our proposed MONO.

Reviewer 3 Report

The authors present SOFIA, a method to optimize multi-target prediction models. They developed the method taking into account of a possible parallelization to improve computational efficiency.

The work is very interesting and the problem of multi-target prediction is well described. Also, the comparison of their MONO with respect to other MOEAs is well done and it clearly shows the advantages of the proposed method.

Anyway, I have some consideration about the implementation of the algorithm:
1. What is the programming language used for algorithm implementation both for CPU and GPU?
2. How this parallelization has been implemented? When the GPU is used, what are the operations carried out by CPU and GPU? Remind that GPU requires SIMD instructions to be correctly used.
3. About the Figures 4a and 4b, I don't understand why the accuracy differs by changing from GPU to 8-threads CPU execution. I expect that accuracy is the same indifferently if it is calculated with GPU or with an undefined number of CPU threads. It should change only execution time.
4. The authors claim to have used an Intel Xeon E7-4830 CPU 2.13GHz CPU. But this CPU has not 64 cores, but 8 cores with a maximum of 16 threads at the same time.

https://ark.intel.com/content/www/us/en/ark/products/53676/intel-xeon-processor-e7-4830-24m-cache-2-13-ghz-6-40-gt-s-intel-qpi.html

Author Response

  1. What is the programming language used for algorithm implementation both
    for CPU and GPU?

The MONO technique is implemented on Java. For the GPU, we have
employed a java framework called TornadoVM. The next information has been
added to the manuscript: “The methods performed in this section have been
implemented in Java. Moreover, MONO and the other methods have also been
run on a GPU with the help of TornadoVM”.

2. How this parallelization has been implemented? When the GPU is used, what
are the operations carried out by CPU and GPU? Remind that GPU requires
SIMD instructions to be correctly used.

The CPU carries out the next operations: (1) initializing the MONO parameters,
(2) finding a random initial solution, (3) assigning the MONO iterations to be

performed for each thread, and (4) presenting the global Pareto solution list. At
the beginning, all threads will have the same initial solution as the current local
solution.
The GPU carries out the next thread operations: (1) obtaining the fitness value
for the current local solution, (2) comparing and updating the global Pareto
solution list, and (3) changing the current local solution according to the MONO
algorithm.
This information have been added in the new version of the manuscript.

3. About the Figures 4a and 4b, I don't understand why the accuracy differs by
changing from GPU to 8-threads CPU execution. I expect that accuracy is the
same indifferently if it is calculated with GPU or with an undefined number of
CPU threads. It should change only execution time.

Approximation algorithms can only find acceptable solutions, not necessarily
optimal. Metaheuristics are a type of approximation algorithms. Because of this,
there is no guarantee that metaheuristics will always get the best results when
exploring the search space. Thus, when changing the number of threads, some
metaheuristic methods obtained better accuracies than with other thread
configurations.

4. The authors claim to have used an Intel Xeon E7-4830 CPU 2.13GHz CPU.
But this CPU has not 64 cores, but 8 cores with a maximum of 16 threads at the
same time.

Thanks for the annotation, we apologize for the mistake. The server has four
Intel(R) Xeon(R) CPU E7- 4830 2.13GHz, each one with 8 cores and 16
threads. So, 32 cores and 64 threads overall. This error has been corrected
within the article.

Round 2

Reviewer 3 Report

The authors answered my questions exhaustively. I have nothing else to add.